# Variations in Surgical Procedures for Inducing Hind Limb Ischemia in Mice and the Impact of These Variations on Neovascularization Assessment

**DOI:** 10.3390/ijms20153704

**Published:** 2019-07-29

**Authors:** Zeen Aref, Margreet R. de Vries, Paul H.A. Quax

**Affiliations:** Department of Surgery, Einthoven Laboratory for Experimental Vascular Medicine, Leiden University Medical Center, 2300 RC Leiden, The Netherlands

**Keywords:** arteriogenesis, angiogenesis, hind limb ischemia, animal model, mouse

## Abstract

Mouse hind limb ischemia is the most common used preclinical model for peripheral arterial disease and critical limb ischemia. This model is used to investigate the mechanisms of neovascularization and to develop new therapeutic agents. The literature shows many variations in the model, including the method of occlusion, the number of occlusions, and the position at which the occlusions are made to induce hind limb ischemia. Furthermore, predefined end points and the histopathological and radiological analysis vary. These differences hamper the correlation of results between different studies. In this review, variations in surgical methods of inducing hind limb ischemia in mice are described, and the consequences of these variations on perfusion restoration and vascular remodeling are discussed. This study aims at providing the reader with a comprehensive overview of the methods so far described, and proposing uniformity in research of hind limb ischemia in a mouse model.

## 1. Introduction

Peripheral arterial disease (PAD) is a major cause of morbidity and mortality [1]. PAD is caused by atherosclerotic plaque progression which leads to the occlusion of peripheral arteries and results in claudication intermittens, or in more severe cases, in critical limb ischemia. The prevalence of PAD increases with age to 20% in people over 70 years [1]. Furthermore, type II diabetes, obesity, and hypertension are known risk factors for atherosclerosis [2]. As increased life expectancy is leading to an increased number of elderly patients suffering from these comorbidities [2], the number of patients with PAD is expected to grow the coming years. Therefore, there is a need for developing new therapies to treat peripheral arterial disease. As a consequence, there is a need for representative models to study and validate these potential new therapies. Therapeutic neovascularization is an alternative therapeutic strategy aimed at improving blood flow to the lower extremities and promoting blood vessel growth. Neovascularization consists of the processes of arteriogenesis, angiogenesis, and vasculogenesis. These processes differ from each other. Arteriogenesis is initiated by shear stress and is the formation of collateral arteries from the pre-existing arteriolar network [3], and will mainly be found in the upper limb of the mice after ligation of the arteries in the limb. The molecular mechanism for arteriogenesis is based on shear stress regulated inflammatory responses, accompanied by the influx of inflammatory cells in the perivascular compartment around the collateral that are being formed.

Angiogenesis is the process of sprouting new capillaries from pre-existing microvasculature. Angiogenesis is mainly driven by ischemia and the upregulation of ischemia-induced transcription factors like HIF1a, and the genes that are responsive to these transcription factors, such as VEGFa and SDF1. In the mouse hind limb model, angiogenesis will mostly occur in the distal parts of the limbs, the gastrocnemius muscle, and the soleus muscle. Vasculogenesis describes the incorporation of circulating (progenitor) cells into the regenerating microvasculature [4], and will only be of relevance in the mouse hind limb ischemia model in cases of cell therapy related studies.

For the purpose of elucidating the cellular and molecular mechanisms underlying the process of neovascularization, and for developing and testing the new therapeutic approaches for neovascularization, a reliable and reproducible animal model is needed.

Animal models of hind limb ischemia have been developed in rabbits, pigs, rats, and mice. A mouse model is preferable over larger animal models because of practical circumstances, and over a rat model because of the wide range of transgenic mice that are available. These are not only mice lacking angiogenic factors like eNOS [5], but also mice deficient in various factors of inflammatory or immune pathways that can subsequently be related to the molecular mechanisms involved in angiogenesis and arteriogenesis [6,7,8,9]. Therefore, the mouse model of hind limb ischemia is the main model used in the preclinical studies. It has to be noted that the mouse model can be performed using different strains of mice, each having a specific pattern of blood flow recovery [10,11,12,13,14]. Applying to all local and governmental regulatory and ethical aspects regarding animal experimentation is, of course, essential.

In this model, limb ischemia is induced by ligation, electrocoagulation, or by applying an ameroid constrictor to occlude the femoral artery. The occlusion of the femoral artery results in arteriogenesis in the thigh and angiogenesis in the distal part of the limb to recover perfusion. Different positions of ligation trigger different pathways of neovascularization. Therefore, the position at which the ligation of the femoral artery should be performed in order to provide a proper model for investigating the perfusion recovery, arteriogenesis, and or angiogenesis remains a topic of research and debate [10]. This illustrates that variations in the model for inducing hind limb ischemia in mice may lead to differences in blood flow recovery and vascular repair or neovascularisation, and may also have consequences for the interpretation of the data obtained in relation to the underlying pathophysiological mechanisms. In the literature, the tourniquet-induced hindlimb ischemia-reperfusion method is also described to induce acute ischemia [15,16,17]. For the current review, this method is less relevant, since the induced ischemia is for such a short period that no effects on neovascularization are observed.

Moreover, as will be discussed below, it is essential to consider which mouse strain to use, because genetic background influences the outcome. Several studies have shown that different mouse strains display a high variability in the degree of neovascularization [13,14]. Many studies have shown fast recovery of limb perfusion in C57BL/6 after inducing hind limb ischemia in comparison to the slow recovering Balb/C mice. Furthermore, it is important to realize that immune-compromised mice such as nude mice or non-obese diabetic-severe combined immunodeficiency (NOD-SCID) mice that are frequently used for cell therapy studies [18] have an increased number of pre-existing collaterals and require a more fierce induction of ischemia, including the removal of arterial branches [18].

The aim of this paper is to review the techniques of inducing hind limb ischemia in a mouse model, providing a rationale with which to identify the optimal mouse model. To this end, we describe variations in anatomical nomenclature, variation in surgical methods, and histopathological and radiological techniques for the quantification of the end-points.

## 2. Mouse Model for Hind Limb Ischemia

The mouse model of hind limb ischemia was first described by Couffinhal et al. [19] based on the rabbit model for hind ischemia, as developed in the group of Wolfgang Schaper. They induced acute hind limb ischemia by ligating the proximal end of the femoral artery, and the distal portion of the saphenous artery, and subsequently excised the femoral artery and all side-branches. The recovery of the blood flow in the limbs was monitored for 5 weeks postoperatively by Laser Doppler Perfusion Imaging (LDPI). The blood flow was reduced postoperatively and the reduction maintained for 7 days; then, the flow was increased over the course of 14 days, reaching a plateau between days 21 and 28 [19]. Thereafter, many groups used the model and also developed new adapted approaches.

### 2.1. Vascular Anatomy of the Mouse

The mouse hind limb is perfused by the external iliac artery (EIA), which changes in the common femoral artery after it passes the inguinal ligament. The femoral artery gives rise to branches that further divide into collateral vessels that penetrate the muscles.

Kochi et al. described in detail the arterial anatomy of the mouse hind limb, the distribution of collaterals, and also three collateral arterial routes [20]. They indicated that there are collateral artery routes through the quadriceps femoris, the biceps femoris muscles, and medial thigh muscles [20]. The medial thigh muscles include the adductor muscles, and are perfused by the proximal caudal femoral artery, as named by Kochi et al.; however, the same artery is termed the deep femoral artery by Limbourg et al. [20,21]. It is the same artery which gives rise to collateral arteries that pervade the adductor muscle group and are the most evaluated muscle groups for arteriogenesis in the tissue after inducing hind limb ischemia.

Kochi et al. highlighted the confusion in the literature about the arterial anatomy due the embedding of the arteries in the musculature and the fact that some of the small arteries are only visualized after dilating and fixing the arteries and, if needed, dissecting the vein to better expose the artery. In order to compare the different approaches used to induce hind limb ischemia, it is important to name the structures identically, and it is more important to describe the procedure and, if possible, to add images of the ligation site to make comparisons possible between the studies and a correct interpretation of the results.

### 2.2. Technical Aspects of Inducing Mouse Limb Ischemia

For the induction of hind limb ischemia in mice, the blood flow in vessels supplying the limb needs to be interrupted by surgical intervention. The general steps of the procedure are described below, whereas the details on the variations in the procedure will be described in the following sections. After anesthetizing a mouse, it is positioned in dorsal decubitus with the hind limbs externally rotated. In one limb, ischemia was operatively induced by ligation or coagulation, with the other limb serving as an internal control. Ligation and coagulation are considered equal in inducing hind limb ischemia, even though coagulation may induce recoil of the femoral artery. For anesthesia two types are commonly used: inhalation, using isoflurane [22], and injectable, using different combinations. The most used combination is midazolam (5 mg·kg^−1^) and medetomidine (0.5 mg·kg^−1^) [10] or a combination of 100 mg/kg of ketamine (100 mg·kg^−1^) and xylazine (10 mg·kg^−1^) [23]. However, alpha-agonists induce early peripheral vasoconstriction and may disturb the results of the experiment [24]. Therefore, ketamine and alpha-agonist combinations are not suitable for studies of vascular smooth muscle cell rich vasculature [23]. For analgesics, fentanyl (0.05 mg·kg^−1^) is most commonly used. For anesthesia reversal, atipamezole (1 mg·kg^−1^) is used. After surgery, the skin can be closed with 6-0 (Ethilon) sutures.

### 2.3. Variants of Surgical Procedure

To induce hind limb ischemia, different surgical approaches have been used (Figure 1, Table 1), ranging from a single ligation of the femoral artery or iliac artery [11,25] to a complete excision of the femoral artery and its side-branches [19], sometimes even in combination with dissection of the vein and the nerve [26]. Also, there is a variation in the level of inducing the arterial occlusion ranging from proximal ligation of the iliac artery to distal ligation just proximal to the bifurcation of the saphenous artery and the popliteal artery.

#### 2.3.1. Single Electrocoagulation or Ligation of Femoral Artery

This surgical method is the most used method to induce hind limb ischemia. Through an incision in the inguinal region, the femoral artery is exposed. Thereafter, the subcutaneous inguinal fat pad is pulled aside and the artery is dissected from the vein and the nerve. Then the femoral artery is ligated or coagulated. The anatomical level at which the occlusion of femoral artery is induced differs among studies. Limbourg et al. ligated the femoral artery distally to the origin of the deep branch [21], which leads to increase of blood flow and shear stress in the collaterals where remodeling occurs. The deep branch of the femoral artery gives rise to collaterals in the adductor muscles, and after ligation of femoral artery distally from it, remodeling will take place and collaterals will develop, as is the case in neovascularization in human PAD. In some studies, the electrocoagulation level is more proximal in the femoral artery. The most important consideration for the site of ligation is that collaterals need to form from proximal side branches of the femoral artery; thus, the ideal site of ligation should be distal of side branches of the artery.

Furthermore, ligation of the femoral artery also induces ischemia and triggers angiogenesis in the distal part of the limb, in the calf muscles, especially in the gastrocnemicus muscle.

#### 2.3.2. Single Electrocoagulation of Iliac Artery

For the exposure of the iliac artery, two approaches are used; Westvik et al. performed a midline incision in the abdomen exposing the common iliac artery and vein and then ligated both, proximally from the origin of the internal iliac artery [26]. Hellingman et al. used the approach of making an incision in the inguinal region and retroperitoneally moving the peritoneum proximally with a cotton swab. Thereafter, the artery is prepared from the vein and the common iliac artery is electrocoagulated proximally from the origin of the internal iliac artery [10,18,27]. In this method arteriogenesis in the upper thigh muscles in all the three collateral compartments can be evaluated. Also, angiogenesis in the distal calf muscle can be evaluated. The recovery of blood perfusion in C57Bl6 mice analyzed by Laser Doppler Perfusion Imaging (LDPI) was complete in 7–14 days. This was the same as the blood reperfusion period in single electrocoagulation of the femoral artery [10].

#### 2.3.3. Double Electrocoagulation of Both Femoral Artery and Iliac Artery

An incision in the inguinal region is made, and through the retroperitoneal approach, the peritoneum is moved proximally with a cotton swab. In this model, both the common iliac artery and femoral artery were electrocoagulated. The common iliac artery is ligated proximally of the internal iliac artery and the femoral artery is electro-coagulated proximal to the superficial epigastric artery [10]. Alternatively, a double electrocoagulation may be performed in the femoral artery, one just distally to the origin of the deep branch and the second one just above the bifurcation of the saphenous and popliteal artery. These double occlusion models are suitable for assessing both arteriogenesis and angiogenesis. The double electrocoagulation leads to severe ischemia and prolonged time course of blood flow recovery in C57Bl6 mice up to 21 days for reaching the plateau in flow recovery. This makes the method suitable for studying new therapeutic approaches, as it offers, due to a higher degree of tissue damage as well as ischemia, a therapeutic window in which improvements can be evaluated.

#### 2.3.4. Total Excision of the Femoral Artery

Total excision of the femoral artery was the first described method by Couffinhal, Isner and colleagues [19]. An incision is made in the skin overlying the middle portion of the hind limb. The common femoral artery is exposed and dissected from the vein and nerve in the distal direction. All side branches of the artery were dissected free and coagulated. The distal ligation level was at the popliteal artery level, just distal from the bifurcation of the saphenous artery and the popliteal artery. After cutting the artery between the two ligatures proximal and distal, the whole artery was removed from the surrounding tissue. The advantage of this method is that a severe ischemia is induced in the calf muscle, and angiogenesis can be assessed. The disadvantage is that the collateral artery formation is impeded due to the disturbance of the pre-existing arterial bed and connections. Therefore, this method is not suitable for assessing arteriogenesis, but is very adequate for assessing angiogenesis in ischemic distal tissue.

#### 2.3.5. Ameroid Constrictors

One of the shortcomings of all the aforementioned models is the fact that all are for acute induction of ischemia in the hind limb, whereas PAD patients mostly suffer from a gradual occlusion of the blood vessels in the legs. Yang et al. and Padgett et al. used ameroid constrictors to induce hind limb ischemia [6,28]. Ameroid constrictors are devices that occlude the artery over 1–3 days through gradually absorption of moisture from the surrounding tissues. By using this method, the blood flow fell to its lowest in 3 days instead of directly after surgery in the traditional hind limb ischemia model. The aim of this method was to develop a more subacute ischemia instead of an acute one. However, Yang et al. showed that this method leads to different responses in the thigh than the traditional ligation model. In the thigh, where the remodeling of collaterals occurs, was a lack of upregulation of shear stress responsive genes and inflammatory genes, which are essential for the process of arteriogenesis, despite the formation of collaterals. There are also technical challenges, namely that commercially-available ameroid constrictors are variable in depth and shape. Furthermore, the severity of the induced ischemia is influenced by the vessel size which depends on age, and age differences of only a few weeks can lead to vessel size differences.

This model has potential. However, for studying the effects of flow recovery, it is hampered by the fact that gradual occlusion leads to a gradual induction of collateral formation to compensate for the occlusions. Consequently, distal ischemia, and thus angiogenesis, is hardly observed. Although this model has to potential to better mimic the situation in patients with chronic ischemia, more research is needed to optimize the method.

## 3. Analysis of Blood Flow Perfusion and Neovascularization/End-Points

Different approaches are used to visualize the blood flow recovery and evaluate the arteriogenic effects and the angiogenic effects.

### 3.1. Laser Doppler Perfusion Imaging

The essential readout of the model is the time course and extent of the blood flow recovery in the ischemic limb. The blood reperfusion in the ischemic limb and control limb is mainly analyzed by Laser Doppler Perfusion Imaging (LDPI). LDPI is based on the principle whereby the Doppler effect caused by the interaction between the laser light and red blood cells is depicted on a color scale. The blood perfusion values are measured per pixel and the region of interest can be indicated on the images by manually drawing these regions of interest. The mean of perfusion in this region is calculated. The ratio of blood flow in the ischemic to the control limb is the general method to express the blood flow reperfusion.

Laser Doppler Perfusion Imaging is a non-invasive method and is reproducible and repeatable under the same experimental conditions at different time points. The perfusion signal is influenced by various experimental conditions including the presence of hair, the type of anesthesia that mice are subjected to and body temperature [23]; it is also associated with movement. Therefore, standardization of the analysis conditions including the environmental temperature is essential. Keeping the animals at physiological temperatures ~37 °C during the Laser Doppler measurements is strongly recommended using e.g., a body temperature control pad or a 37 °C double glass water bowl.

For measuring the recovery of the blood flow, different regions of interest can be used. These regions vary from whole limb with or without the inguinal region, or the footpad. A critical point in whole limb analysis is that increased angiogenesis due to wound healing at the site of the incision should be taken into account. The footpad is the most reliable region, since it is hairless, and shaving is not required. Shaving can irritate the skin, and therefore, influences the flow signal. The hair absorbs laser light and prevents it from interacting with red blood cells. Other methods for removing the hair such as the use of hair removal creams can be considered, but usually require that the mouse be under anesthesia for a longer period, and can irritate the skin too.

The LPDI analysis is generally performed at predetermined time points; before the surgery, directly after the surgery, after 3 and 7 days, and thereafter weekly over a period of 2 or 4 weeks.

### 3.2. Immunohistochemical Analysis

Next to evaluating the blood flow restoration, usually a histological analysis is performed to study arteriogenic and angiogenic responses in the tissue. Immunohistochemical analysis can be used to detect neovessels, as well as inflammatory cells that infiltrate the tissue. In addition, histological analysis can also be used for analysis of skeletal muscle remodeling, damage, and necrosis.

Arteriogenesis is studied in the proximal part of the limb, in the adductor muscle group. The collateral formation is determined by immunohistochemical staining using antibodies against alpha-Smooth Muscle Actin (a-SMA) to demonstrate the arterial nature of the newly formed vessels in the adductor muscle [7].

For assessing angiogenic capillary density in the distal part of the limb, the ischemic calf muscle, is most commonly used. The angiogenic capillaries are detected by using immunohistochemical techniques to identify endothelial cells (e.g., CD31 or von Willebrand factor) [29,30].

### 3.3. Other Methods for Assessment of Collateral Formation and Limb Perfusion

For assessments of vascular remodeling, both the evaluation of the anatomic dimensions and the functionality of the newly formed vessels are of interest. Methods for evaluations will be discussed in the following section and are summarized in Table 2.

The traditionally-used X-ray microangiography after administration of an iodinated contrast agent can provide a gross anatomical view of the circulation visualizing the vessels in the range of 25 to 50 μm in diameter. This method is invasive and provides only 2-D projection images. Furthermore, the need for anesthesia and vascular access makes this technique technically challenging.

A different method is to inject the aorta with polyacrylamide-bismuth contrast inducing vasodilation followed by post-mortem angiographic images to study the collateral vessel growth [31]. Although technically less demanding, the obvious disadvantage of this method is that it is acquired post-mortem, and therefore, repeated measurements are not possible.

For the anatomical visualization of the arterial circulation, high-resolution micro-computed tomography (Micro-CT) is the best method, which is reproducible, and the 2D images can be reconstructed as 3D images [32]. Also, for the quantification of the extent of arteriogenesis, micro-CT can be used by analyzing the obtained images for the number, volume and length of the newly-formed collateral arteries [33]. Using Micro-CT in combination with the administration of intra-arterial contrast medium makes it possible to visualize vessels of 8 μm in diameter [34]. The challenge of this technique is discriminating arteries from veins after administration of the contrast agent. Also, underfilling the arteries with the contrast agent, or conversely, overfilling leading to extravasation of the contrast agent, are possible problems impairing comparisons between different measurements.

Another method is casting the vasculature with a silicone radiopaque casting agent (Microfill) and post-mortem imaging using Micro-CT to quantify changes in the microvasculature [35].

For the functional assessment of blood flow recovery and tissue perfusion, magnetic resonance imaging (MRI) is a promising technique to detect arterial blood flow and to follow the collateral arterial formation after occlusion of the femoral artery in the hind limb ischemia model. There are two MR imaging techniques, namely contrast-enhanced MRA with a gadolinium-based contrast agent and the time-of-flight (TOF) sequence technique. TOF is an MRI technique to visualize flow within vessels, and is based on flow-related increase of spins entering into a single image slice; as a result of being unsaturated, these spins of the blood give more signal than the stationary spins of the surrounding tissue [36,37]. Wagner et al. showed that the use of MRI angiograms is an effective technique for determining the collateral formation, and whether the collateral arteries in the quadriceps muscles are better developed than those in the adductor muscle group [38]. In another study, Wagner et al. showed that TOF MRI imaging without using a contrast agent is effective in mice, and can be used for quantitative evaluations of arterial blood flow. The advantage of this method is that it does not require a contrast agent, and therefore, no vascular access for injection of contrast is needed; furthermore, repetitive analyses on one animal is feasible, without considering the residue of the used contrast agent [37]. Using a non-invasive MRI technique to follow the development of collateral arteries is a promising technique, now even more so as the images can be combined using a reconstruction technique of maximum intensity projection (MIP) to acquire a 3D image of the vessels, similarly to conventional angiography.

Hendrikx et al. showed that single photon emission computed tomography (SPECT) perfusion imaging using radioisotope-based tracers can be used to analyze perfusion recovery and muscular damage in the mouse hind limb model [39]. This nuclear perfusion imaging and myocyte damage imaging are non-invasive and have high resolution. The results of the study demonstrated that LDPI analysis underestimates the revascularization processes in the hind limb ischemia model. When interpreting the results of LDPI, researchers should take into account that the main blood flow recovery occurs in the first 7 days. The disadvantage of this method is that it needs special facilities to work with the radioisotope-based tracers.

Microsphere injection and contrast enhanced ultrasounds are two further techniques currently in early experimental phases of research.

### 3.4. Methods to Further Differentiate the Results of HLI and Neovascularisation

#### 3.4.1. Matrigel Plug Assay

Studies using the mouse hind limb ischemia model are often directed at the discovery of factors that improve or impair blood flow recovery. As mentioned, the blood flow recovery is dependent on the processes of arteriogenesis, angiogenesis, or both. A method to further differentiate whether the blood recovery is due arteriogenesis or angiogenesis is performing an in vivo Matrigel plug assay.

The in vivo Matrigel plug assay is an assay solely intended to assess angiogenesis; it is performed by injecting Matrigel into the subcutaneous space on the dorsal side of the mice on both the left and right flank [7]. Post-implantation, after 7 to 14 days, the mice are sacrificed and the Matrigel plugs are excised and processed for immunohistological analysis. Paraffin sections could be stained with a general staining such as Hematoxylin, Eosin or Hematoxylin, Phloxine and Saffron, and anti-CD31. The CD31 staining affirms the endothelial nature of the infiltrating capillary structures. The extent of the angiogenesis is determined by assessing the vascular ingrowth in the Matrigel plug. The vascular ingrowth is scored by measuring the depth of ingrowth and the length of the capillary structures, and provides information on the angiogenic potential of the condition or compound being studied.

#### 3.4.2. Pre-existing Collateral Density

The dominant mechanism responsible for the restoration of the blood flow after arterial occlusion is the remodeling of the pre-existing collateral arterioles into mature functioning collateral arteries.

There are pre-existing interarterial collateral connections in the peripheral circulation. The amount of these pre-existing arterioles varies between different strains, and different factors can influence the pre-existing vascular bed. Pre-existing collateral density can be determined in pial circulation of the pia mater [40]. The pre-existing collateral density in the cerebral pial circulation gives an indication of collateral density in skeletal muscle [8].

To determine the pial circulation in mice, after anesthesia, the thoracic aorta is cannulated retrograde, and the circulation is maximally dilated by infusion of sodium-nitroprusside and papaverine in PBS at approximately 100 mmHg prior to vascular casting. After craniotomy, Microfil is infused under a stereomicroscope. The dorsal cerebral circulation is then fixed by the topical application of 4% paraformaldehyde (PFA) to prevent any degradation in vessel dimensions after Microfil injection. The brains are fixed overnight in 4% PFA, and subsequently incubated for several days to improve the contrast of the visualization of the vasculature. The number of pre-existing collaterals in the semi-hemispheres of the pia mater is subsequently quantified, and can be used as a general degree for pre-existing arterioles in the particular strain of mice [14,41].

## 4. Discussion

The mouse hind limb ischemia model is the most used model for basic research and for pre-clinical studies investigating therapeutic neovascularization. It is considered a proper model, as it is reproducible, and strongly mimics the specific features of human peripheral arterial disease.

To perform this model, occlusions at different anatomical levels of the iliac and femoral artery are used, as well as different technical approaches. Based on the knowledge that arteriogenesis can only occur after increasing shear stress in the pre-existing arterioles proximal to the occlusion, the correct level of ligation of the femoral artery is distal to the origin of the collateral branches. However, in methods in which the femoral artery is ligated proximal to the origin of the collateral branches, arteriogenesis is also observed. Kochi et al. demonstrated that the number of collaterals is higher than previously thought. This may explain the unexpected arteriogenesis after ligation of the femoral artery at the proximal end. Kochi et al. also demonstrated the presence of collateral arteries in other muscle groups in the upper thigh besides the adductor muscle group. Furthermore, recent MRI research demonstrated that the collateral arteries in the quadriceps muscles are better developed than those in the adductor muscle group. This suggests that the analysis of the other muscle groups may yield valuable information on arteriogenesis, and should be included in experimental studies.

The hind limb ischemia mouse model has limitations. One of the major limitations is the acute nature of the ischemia induced in this model, while the PAD in patients is a chronic process, and critical limb ischemia arises slowly as a result of a gradual build-up of atherosclerosis. Yang et al. [28] and Padgett et al. [6] used ameroid constrictors to progressively induce hind limb ischemia and mimic the gradual occlusion that occurs in patients. However, this method is not yet representative of the cellular response in the thigh after occlusion in PAD patients; therefore, further optimization of the method is needed.

Another limitation related to the hind limb ischemia model is that the procedure is predominately performed in young and healthy mice, which do not reflect patients with PAD. PAD patients are old and have co-morbidities like diabetes mellitus, hypercholesterolaemia, and hypertension. Westvik et al. showed that old mice show a slower perfusion recovery after inducing hind limb ischemia compared to young mice [26]. Young wild-type mice do not show the comorbidities of PAD patients. It is well known that these comorbidities accelerate the development of atherosclerosis and affect vascular remodeling [12]. The extent to which these comorbidities affect vascular remodeling varies. Van Weel et al. investigated arteriogenesis using a hind limb ischemia model in different mouse types and showed that hypercholesterolaemia is more than hyperglycemia or hyperinsulinemia associated with impaired arteriogenesis [12]. Hypercholesterolaemia leads to impaired arteriogenesis due elevated blood cholesterol levels which affect monocyte chemotaxis, whereby monocyte influx is reduced [42]. These data suggest that the changed lipid metabolism in diabetes patients may affect the arteriogenesis more than a disturbed glucose metabolism. Moreover, diabetes mellitus causes endothelial dysfunction which is multifactorial and results in reduced angiogenesis [43].

Hypertension activates angiotensin, which is associated with endothelial dysfunction due increased oxidative stress. However, Angiotensin activation can initiate and stimulate arteriogenesis due to inflammation regulation. Hypertension can also stimulate arteriogenesis through the increase of shear stress and activation of the renin-angiotensin system [42].

The mouse hind limb ischemia model and PAD patients frequently show similar neovascularization patterns with arteriogenic responses that are close to the occlusions to form collaterals arteries and an angiogenic response in the distal ischemic tissue, as was illustrated by comparing the angiogenic response and VEGF expression in the muscle biopsies of CLI patients and mice after induction of HLI [31,44,45]. However, there may also be differences in neovascularization patterns observed. In the mouse HLI model, neovascularization occurs in a standard, homogenous fashion, due to the fairly standard way of blocking the blood flow, i.e., by occluding the major arteries in the upper limb in an acute way. This may be a crucial difference with the situation in PAD patients, where occlusion usually occurs gradually, but most importantly, it may occur at various locations in the vascular tree. It is obvious that an occlusion of one of the major arteries above the knee may have different consequences on endogenous collateral formation and neovascularization than an occlusion below the knee. Therefore, the neovascularization required to restore the blood flow in PAD patients with critical limb ischemia will be quite heterogeneous in nature, whereas the mouse models usually focus on a more standardized neovascularization induction. Unfortunately, mimicking the human situation is still complex, and really predictive (larger) animal models are not available yet, making the mouse model the most used model at present.

Furthermore, to choose an appropriate model, it is essential to consider which mouse strain to use, because the genetic background influences the outcomes. Several studies have shown high variability between mouse strains in restoring the limb perfusion after the induction of ischemia, and variability in ability of neovascularization [13,14]. Different strains show different blood recovery patterns after surgically-induced hind limb ischemia. Many studies have exhibited fast recovery of limb perfusion in C57BL/6 after inducing hind limb ischemia in comparison to the slow recovery of Balb/C mice. It has been suggested that the difference in vascular remodeling is due a specific gene locus in chromosome 7 of the mouse [46,47]. Also, a wide variation in the extent of the native pre-existing collaterals is observed in different mouse strains, whereby C57BL/6 and BALB/c demonstrate the largest difference [40]. Due to their slow recovery, Balb/c mice are often used for hind limb ischemia studies, based on the assumption that this slow response better mimics the situation in patients with PAD. Recently Nossent et al. showed that especially in Balb/c mice, a stronger upregulation of pro-angiogenic and pro-arteriogenic genes is observed when compared to C57Bl6 mice, despite the poorer blood perfusion recovery in Balb/c [7]. This suggests Balb/c mice lack a thus far unknown factor that is crucial for vascular remodeling, rather than that this model better mimics the situation in patients with peripheral arterial disease.

Schmidt et al. recently demonstrated more muscle injury within 6 h after inducing ischemia in Balb/c mice compared to C57BL/6 mice [48]. The muscle injury may contribute to the ability of the vascular bed to recover after ischemia, and to regenerate. This suggests that understanding of etiology of the ischemic muscle and its contribution to the restoration of blood flow is needed, and that more research on this topic is required.

The techniques for the assessment of the results of the hind limb ischemia model are diverse. However, LDPI remains the essential readout method because of feasibility and reproducibility. The other techniques can serve as additional techniques since they mostly lack the capability to perform robust and fast analysis of the blood flow in larger series of mice.

## 5. Conclusions

The mouse hind limb ischemia model is performed in many variants. No single method for inducing hind limb ischemia that is appropriate for all research questions. The researchers should recognize the variants to be able to choose the most suitable model for their experiments, and they should describe the method and the used anatomical landmarks in order to make comparisons possible among studies.

## Figures and Tables

**Figure 1 ijms-20-03704-f001:**
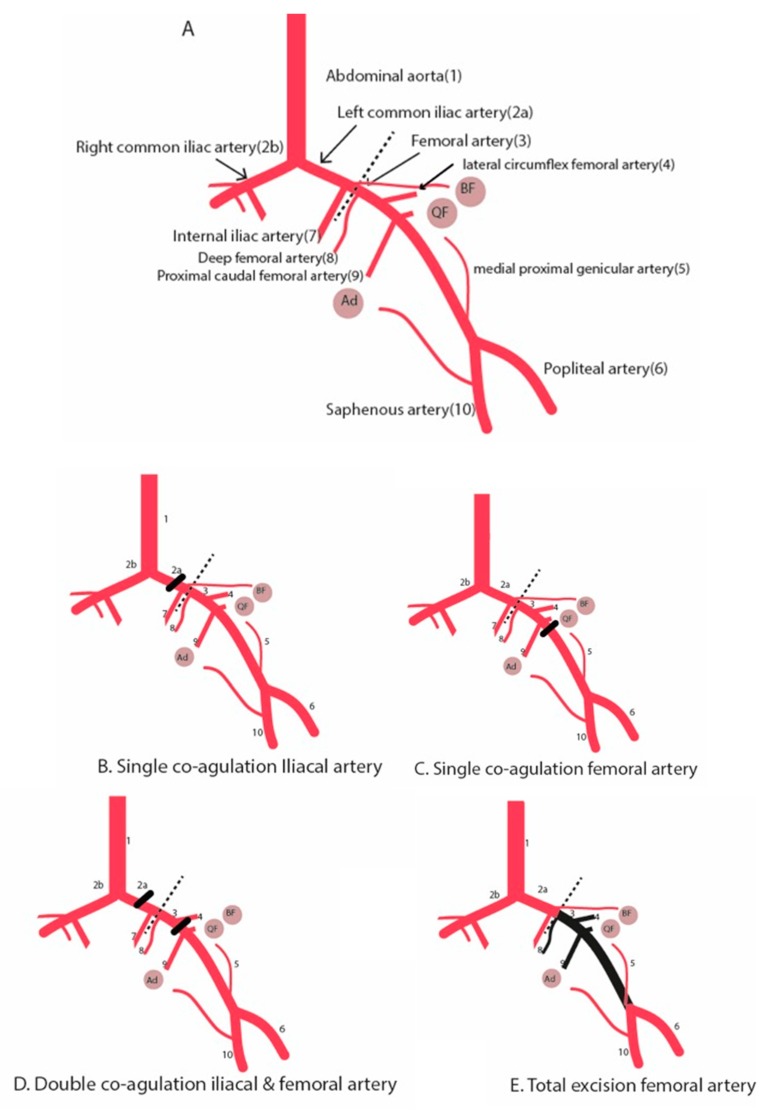
Schematic diagram illustrating the hind limb ischemia and different surgical methods to induce hind limb ischemia. (**A**) Hind limb vasculature. (**B**) Single electrocoagulation of the iliac artery. (**C**) Single electrocoagulation of the femoral artery. (**D**) Double electrocoagulation of the iliac and femoral artery. Alternatively, the electrocoagulation of the iliac artery can be replaced by a second electrocoagulation of the femoral artery just above the bifurcation of the saphenous and popliteal artery. (**E**) Total excision of the femoral artery. Round circles are muscle groups: (Ad) adductor muscle group. (QF) Quadriceps femoris. (BF) Biceps femoris. Dotted line is the inguinal ligament. Black beam is the occlusion site.

**Table 1 ijms-20-03704-t001:** Surgical methods used to induce hind limb ischemia, time course of blood reperfusion recovery, whether the method is suitable to assess arteriogenesis and angiogenesis, and if arteriogenesis takes place, where it will take place.

Surgical Method Used to Induce HLI	Time Course of Blood Reperfusion (Days)	Ischemia Rate	Suitable to Studying Arteriogenesis?	Which Collateral Group Can Be Studied?	Suitable to Studying Angiogenesis in Calf Muscle?
Electrocoagulation of common iliac artery	7–14 days	moderate	Yes	Medial thigh muscles (includes adductor muscles) Quadriceps femoris Biceps femoris	Yes
Electrocoagulation of the femoral artery, distal from origin of deep branch	7–14 days	mild	Yes	Adductor muscles Quadriceps femoris Biceps femoris	Yes
Electrocoagulation of the femoral artery, proximal from origin of deep branch	7–14 days	mild	Yes	Quadriceps femoris Biceps femoris	
Total excision of the femoral artery	28 days	severe	No		Yes
Double electrocoagulation of iliac and femoral artery	28 days	moderate	Yes	Medial thigh muscles (includes adductor muscles) Quadriceps femoris Biceps femoris	Yes
Ameriod constrictors		mild	Unclear		No

**Table 2 ijms-20-03704-t002:** Methods of assessment of collateral formation and limb perfusion.

Techniques	Results Obtained	Advantages	Disadvantages
Micro-CT	- Anatomical visualization of vasculature circulation - Extent of vasculature formation	- Non-invasive - Reproducible - Repeatable - 3D reconstruction	- Challenging to discriminate arteries from veins - Ionizing radiation
Ex vivo Micro-CT of polymer casted vasculature	- Extent of vasculature formation - Quantification of changes in the microvasculature	- Only arteries - 3D reconstruction	- Invasive - Repeated measurement not possible
X-ray microangiography	- Gross anatomical view of the circulation	- Overview of the vascular anatomy	- Invasive - Technically challenging - Only 2D projection images
Post-mortem X-ray microangiography	- Gross anatomical view of the circulation	- Overview of the vascular anatomy	- Repeated measurement not possible
MRA	- Detecting arterial blood flow - Determine collateral formation - Follow collateral arterial formation	- Repeated measurements possible - 3D reconstructions	- Invasive (vascular access for injecting contrast) - Long scan times
MRI TOF	- Visualizing flow within vessels - Quantitative evaluation of arterial blood flow - Determine collateral formation	- Non-invasive - Repeated measurements possible - 3D reconstructions	- Long scan times
SPECT	-Analyzing perfusion recovery	-Non-invasive -Accurate	-Special facility is needed

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
