# Peer review of "Variations in Surgical Procedures for Inducing Hind Limb Ischemia in Mice and the Impact of These Variations on Neovascularization Assessment"

_ijms, 2019, doi:10.3390/ijms20153704_

Round 1

Reviewer 1 Report

This Review manuscript by Aref et al provides a comprehensive and up to date revision of the methodologies used for the hing limb ischemia experimental model. It is well written, easy to read, and adds to the scientific literature by highlighting the need for harmonizing methodologies to increase reproducibility. Below some comments for the authors to address.

The potential to assess arteriogenesis and angiogenesis varies depending on the model used. This is clearly described and summarise in table 1; however I would suggest to add to the simple YES/NO, a subjective/qualitative assessment of the level of ischemia generated in tissues. This also relates to the comment that the double electrocoagulation has a bigger therapeutic window, but probably the level of ischemia and tissue damage is larger.

Authors present many variables that impact on the laser doppler readouts; however they do not provide their views on the optimal protocols. To facilitate harmonization, could the authors based on their experience suggest the optimal temperature to perform the doppler and if hair removal is needed?

Discussion includes a critical point about the mouse strains. This should be in the body of the manuscript and authors should discuss something about immunodeficient mice.

It is important that authors comment on the need to align with regulatory policies for animal welfare and also the need for appropriate ethical approvals.

Authors state that the use of alpha agonist is not suitable, which is important; but reference for this statement seems not appropriate.

Please correct multiple minor typographical errors.

Author Response

Comments and Suggestions for Authors

REVIEWER 1:

This Review manuscript by Aref et al provides a comprehensive and up to date revision of the methodologies used for the hing limb ischemia experimental model. It is well written, easy to read, and adds to the scientific literature by highlighting the need for harmonizing methodologies to increase reproducibility. Below some comments for the authors to address.

AUTHORS: 

We thank the reviewer for the careful and constructive analysis

REVIEWER 1:

The potential to assess arteriogenesis and angiogenesis varies depending on the model used. This is clearly described and summarise in table 1; however I would suggest to add to the simple YES/NO, a subjective/qualitative assessment of the level of ischemia generated in tissues. This also relates to the comment that the double electrocoagulation has a bigger therapeutic window, but probably the level of ischemia and tissue damage is larger.

AUTHORS: We thank the reviewer for this helpful suggestion and table 1 is adapted according the suggestion of the reviewer. Indeed, the double electrocoagulation has a bigger therapeutic window, due to a higher degree of tissue damage as well as ischemia. We have clarified this point in the text (see page 6, last sentence of that section)

REVIEWER 1:

Authors present many variables that impact on the laser doppler readouts; however they do not provide their views on the optimal protocols. To facilitate harmonization, could the authors based on their experience suggest the optimal temperature to perform the doppler and if hair removal is needed?

AUTHORS; according the reviewer’s suggestion we have adapted this part and stressed the importance of heat control during the LDPI measurement. Also, we added some details about the technical aspects of the procedure. However, we believe that providing a procedural protocol here would hamper the readability of the manuscript.

REVIEWER 1:

Discussion includes a critical point about the mouse strains. This should be in the body of the manuscript and authors should discuss something about immunodeficient mice.

AUTHORS: We agree with the reviewer that this is an important point and address this aspect already partly in the introduction of the manuscript (last section of the introduction). Furthermore, we adapted this section, including a new section on immunodeficient mice. In the discussion we left the detailed discussion on the strain differences.

REVIEWER 1:

It is important that authors comment on the need to align with regulatory policies for animal welfare and also the need for appropriate ethical approvals.

AUTHORS: We fully agree with the reviewer and a sentence was added to stress this important aspect (page 3, 3rdparagraph last sentence: “ Applying to all local and governmental regulatory and ethical aspect regarding animal experimentation is of course essential.” )

REVIEWER 1: authors state that the use of alpha agonist is not suitable, which is important; but reference for this statement seems not appropriate.

AUTHORS: We thank the reviewer for this remark and added the proper reference.

REVIEWER 1: Please correct multiple minor typographical errors.

AUTHORS: We apologize and checked the manuscript thoroughly for typos

Reviewer 2 Report

Reviewing the manuscript entitled, “Variation in surgical procedures for inducing hind limb ischemia in the mouse and the impact of these variations on neovascularization assessment.” by Aref Z. et al., overall, this is a clear, concise, and well-written review manuscript about hind limb ischemia methods in mouse. However, the explanation of the overall course of contents in the introduction and the content of each reviewing are not consistent. Therefore, the authors need to response my concerns for accepting.

Concerns

#1.In introduction, authors start from description of PAD. However, after that, there is little mention about PAD until limitation. I believe the essence of this manuscript is the sentence described on line 68 page3 in introduction. Therefore, authors need to describe the relationship to the present situation between PAD therapy and mouse animal models.

#2. On line 77 in in introduction, authors mentioned importance of elucidation of molecular mechanisms of PAD. Although authors mentioned surgical techniques in detail, there was little description of changes for the molecular factors caused by each surgery. Authors need to mention the relationship between alternation of the molecular factors and each surgery.   

#3, Description of arteriogenesis, angiogenesis and vasculogenesis is insufficient.

Authors need to mention relationship between vascular repair and vascular genesis, molecular mechanisms of each genesis, the relevance between clinical PAD and each genesis.

#4, Description of advantage in mouse animal model is insufficient. On line 83, Authors interpreted that the wide range of transgenic mice are available. However, there was no description of achievement using the genetically modified mouse in hind limb ischemia surgery. Authors need to explain advantage of mouse model including achievement with the genetically modified mouse.

#5, From line 90 to line 92, several times authors employed “Level”. This is ambiguous. Does it mean ligation position? or strength of ligation? or both?

Authors need to explain this accurately.

Author Response

REVIEWER 2:

Reviewing the manuscript entitled, “Variation in surgical procedures for inducing hind limb ischemia in the mouse and the impact of these variations on neovascularization assessment.” by Aref Z. et al., overall, this is a clear, concise, and well-written review manuscript about hind limb ischemia methods in mouse. However, the explanation of the overall course of contents in the introduction and the content of each reviewing are not consistent. Therefore, the authors need to response my concerns for accepting. 

AUTHORS: 

We thank the reviewer for the careful and constructive analysis

Concerns

REVIEWER 2

#1.In introduction, authors start from description of PAD. However, after that, there is little mention about PAD until limitation. I believe the essence of this manuscript is the sentence described on line 68 page3 in introduction. Therefore, authors need to describe the relationship to the present situation between PAD therapy and mouse animal models. 

AUTHORS :  

To address the reviewers concern we added the following sentence  “ And as a consequence there is a need for representative models to study and validate these potential new therapies.” 

REVIEWER 2

#2. On line 77 in in introduction, authors mentioned importance of elucidation of molecular mechanisms of PAD. Although authors mentioned surgical techniques in detail, there was little description of changes for the molecular factors caused by each surgery. Authors need to mention the relationship between alternation of the molecular factors and each surgery.   

AUTHORS:

We thank the reviewer for the suggestion and have clarified the link of these crucial aspects of neovascularization in PAD with their role in the mouse hind limb ischemia model in the first paragraph of the introduction. (See also our answer to point #3)

REVIEWER 2

#3, Description of arteriogenesis, angiogenesis and vasculogenesis is insufficient.

Authors need to mention relationship between vascular repair and vascular genesis, molecular mechanisms of each genesis, the relevance between clinical PAD and each genesis. 

AUTHORS:

We thank the reviewer for the suggestion and have clarified the link of these crucial aspects of neovascularization in PAD with their role in the mouse hind limb ischemia model in the first paragraph of the introduction. (See also our answer to point #2)

REVIEWER 2

#4, Description of advantage in mouse animal model is insufficient. On line 83, Authors interpreted that the wide range of transgenic mice are available. However, there was no description of achievement using the genetically modified mouse in hind limb ischemia surgery. Authors need to explain advantage of mouse model including achievement with the genetically modified mouse.

AUTHORS:

We thank the reviewer for this suggestion, we have added a small section on the benefits of using genetically modified mice to unravel the molecular mechanisms in neovascularization. However, we feel that expanding this section to far would deviate to far from the purpose of this manuscript, i.e. providing a review over the technical aspect of inducing hind limb ischemia in mice.

REVIEWER 2

#5, From line 90 to line 92, several times authors employed “Level”. This is ambiguous. Does it mean ligation position? or strength of ligation? or both? 

Authors need to explain this accurately. 

AUTHORS:

We have clarified this point and rephrased level to ligation position

Reviewer 3 Report

The authors have reviewed the various surgical techniques used to induce limb ischemia in mice. It is a well written review. The authors should address the following two issues.

The authors should mention the tourniquet induced hindlimb ischemia-reperfusion method. This method has been used by various groups and can be used to study ischemia induced angiogenesis/arteriogenesis.

The Figure 1 should be drawn to scale to represent the relative size differences between the aorta and the common iliac arteries and the branches. The authors could also draw only the right common iliac artery on the contralateral side to make it easier to understand for readers not conversant with mouse anatomy.

Author Response

REVIEWER 3:

The authors have reviewed the various surgical techniques used to induce limb ischemia in mice. It is a well written review. The authors should address the following two issues.

AUTHORS: 

We thank the reviewer for the careful and constructive analysis

REVIEWER 3:

The authors should mention the tourniquet induced hindlimb ischemia-reperfusion method. This method has been used by various groups and can be used to study ischemia induced angiogenesis/arteriogenesis. 

AUTHORS :

We have added a section on the tourniquet induced hind limb ischemia-reperfusion method to introduction of the manuscript, but we disagree with the reviewer that this model can be used to study effects on neovascularization, due to the short periods of ischemia. This model is more adequate to study the role of ischemia reperfusion damage.

We added the following section to the introduction, lines 112-115 :

“In literature also the tourniquet induced hindlimb ischemia-reperfusion method is described to induce acute ischemia but for the current review this method is less relevant since the ischemia is on induced for such a short period that no effects on neovascularization are observed.” 

REVIEWER 3:

The Figure 1 should be drawn to scale to represent the relative size differences between the aorta and the common iliac arteries and the branches. The authors could also draw only the right common iliac artery on the contralateral side to make it easier to understand for readers not conversant with mouse anatomy. 

AUTHORS :

We thank the reviewer for this suggestion and we have adapted the figure accordingly.

Round 2

Reviewer 2 Report

Reviewing the manuscript entitled, “Variation in surgical procedures for inducing hind limb ischemia in the mouse and the impact of these variations on neovascularization assessment.” by Aref Z. et al., overall, the authors responded according to my concerns, but partially still insufficient. Therefore, the authors need to response my concerns for accepting.

Concerns

#1.Last time, I asked what is the relationship between alternation of the molecular factors and each surgery. In the revised version, I think that your response indicates “blue words” in line 79 to 85. I understand that anatomical position of ligation in the limbs is very important for decision of arteriogenesis or angiogenesis. Is this correct? However, my question is what is the difference in molecular mechanisms between arteriogenesis and angiogenesis? Since these animal experimental models are necessary for understanding the future therapy of human PAD as mentioned at line 72 page 3, most important factor is the molecular mechanisms in both neovascularization. What kinds of MET are there in PAD? If you do not know the mechanisms, you cannot apply to them for understanding the treatment. Therefore, the authors need to describe the molecular mechanisms in each neovascularization in detail.

#2. I understand that the mouse hind ischemia is the most used model for basic research. However, it is difficult to understand how is the relationship between the neovascularization pattern in the mouse experimental model and human PAD treatment. Can you describe it concretely?

You changed Level to Position. I understood it.

Although I did not point out last time, PAD and peripheral artery disease are mixed in the manuscript. You need to modify it.

Author Response

Reviewing the manuscript entitled, “Variation in surgical procedures for inducing hind limb ischemia in the mouse and the impact of these variations on neovascularization assessment.” by Aref Z. et al., overall, the authors responded according to my concerns, but partially still insufficient. Therefore, the authors need to response my concerns for accepting.

Concerns

Reviewer:

#1.Last time, I asked what is the relationship between alternation of the molecular factors and each surgery. In the revised version, I think that your response indicates “blue words” in line 79 to 85. I understand that anatomical position of ligation in the limbs is very important for decision of arteriogenesis or angiogenesis. Is this correct?

AUTHORS : Yes , that is correct

Reviewer:

However, my question is what is the difference in molecular mechanisms between arteriogenesis and angiogenesis? Since these animal experimental models are necessary for understanding the future therapy of human PAD as mentioned at line 72 page 3, most important factor is the molecular mechanisms in both neovascularization. What kinds of MET are there in PAD? If you do not know the mechanisms, you cannot apply to them for understanding the treatment. Therefore, the authors need to describe the molecular mechanisms in each neovascularization in detail. 

AUTHORS : 

We apologize for interpreting the question in different way the reviewer intended. The reviewer is absolutely correct that the understanding of the molecular mechanism for arteriogenesis and angiogenesis is essential for defining and validating new therapies in these mouse models. The molecular mechanism for arteriogenesis is based on the shear stress regulated inflammatory responses, accompanied by the influx of inflammatory cells in the perivascular compartment around the collateral that are being formed, whereas angiogenesis is mainly driven by ischemia and upregulation of ischemia-induced transcription factors like HIF1a, and the genes that are responsive to these transcription factors such as VEGFa and SDF1. 

We have adapted the text of the introduction to address this point : 

Line 79-85: 

………The molecular mechanism for arteriogenesis is based on the shear stress regulated inflammatory responses, accompanied by the influx of inflammatory cells in the perivascular compartment around the collateral that are being formed.

Angiogenesis is the process of sprouting new capillaries from pre-existing microvasculature. Angiogenesis is mainly driven by ischemia and upregulation of ischemia-induced transcription factors like HIF1a, and the genes that are responsive to these transcription factors such as VEGFa and SDF1. ……….

Reviewer : 

#2. I understand that the mouse hind ischemia is the most used model for basic research. However, it is difficult to understand how is the relationship between the neovascularization pattern in the mouse experimental model and human PAD treatment. Can you describe it concretely?

AUTHORS :

The reviewer touches a critical point that has hampered the progress in the field substantially and which we tried to address in the section on the limitations of the study (line 462-488), but obviously this could be addressed more clearly. We now added a section referring to this point to the discussion on page 16 , line 488 and further.

“The mouse hind limb ischemia model and PAD patients frequently show similar neovascularization patterns with an arteriogenic response close to the occlusions to form collaterals arteries and an angiogenic response in the distal ischemic tissue as was illustrated by comparing the angiogenic response and VEGF expression in the muscle biopsies of CLI patients and mice after induction of HLI[31,44,45].However, there may also be differences in neovascularization patterns observed. In the mouse HLI model the neovascularization occurs in a standard, homogenous  fashion, due to the fairly standard way of blocking the blood flow, i.e. by occluding in an acute way the major arteries in the upper limb. This may be a crucial difference with the situation in PAD patients where occlusion usually occur gradually, but most importantly, may occur at various locations in the vascular tree. It is obvious that an occlusion of one of the major arteries above the knee will have different consequences on endogenous collateral formation and neovascularization than an occlusion below the knee may have. Therefore, the neovascularization required to restore the blood flow in PAD patients with critical limb ischemia will be quite heterogeneous in nature, whereas the mouse models usually focus on a more standardized neovascularization induction. Unfortunately, mimicking the human situation is still complex, and really predictive (larger) animal models are not available yet, making the mouse model still the most used model. “

Reviewer :

You changed Level to Position. I understood it. 

AUTHORS : Thanks

Reviewer : Although I did not point out last time, PAD and peripheral artery disease are mixed in the manuscript. You need to modify it.

AUTHORS : We did an extensive search and replace “peripheral arterial disease” by PAD throughout the manuscript